# Use of Two PGPB Strains for the Valorization of Wastewater Sludge and Formulation of a Biofertilizer for the Recovery of *Quercus ilex*

**DOI:** 10.3390/life15091490

**Published:** 2025-09-22

**Authors:** Vanesa M. Fernández-Pastrana, Daniel González-Reguero, Marina Robas-Mora, Diana Penalba-Iglesias, María José Pozuelo de Felipe, Agustín Probanza, Pedro Jiménez-Gómez

**Affiliations:** Department of Pharmaceutical and Health Sciences, San Pablo University, Urbanización Montepríncipe, 28660 Boadilla del Monte, Spain; vanesa.fernandezpastrana@ceu.es (V.M.F.-P.); diana.penalbaiglesias@ceu.es (D.P.-I.); a.probanza@ceu.es (A.P.); pedro.jimenezgomez@ceu.es (P.J.-G.)

**Keywords:** biofertilizers, metabolic diversity, *Quercus ilex*, metagenomics, PGPB, antibiotic resistance

## Abstract

In response to the growing demand for agricultural production and the need for more sustainable practices, the use of biofertilizers based on the valorization of agricultural residues is presented as an alternative to traditional chemical fertilizers. This approach seeks to minimize environmental impact and improve soil health in agroforestry systems. The present work studies the effect of two plant growth-promoting bacterial strains (PGPB), *Bacillus pretiosus* (C1) and *Pseudomonas agronomica* (C2) on *Quercus ilex* (holm oak) seedlings. Taxonomic diversity was evaluated by massive sequencing of amplicons of the 16S rRNA gene, as well as the metabolic diversity and antibiotic resistance profile of the bacterial communities. The study also evaluated the impact of PGPB strains on the development of *Quercus ilex* seedlings. On the other hand, the effect of the biofertilizer on soil bacterial communities was evaluated. The results showed that the addition of biofertilizer significantly improved plant development compared to the addition of traditional irrigation (water) or the addition of fertilizer without the strains. In the same way, it was observed how the addition of the strains reduced the minimum inhibitory concentrations (MIC) in the rhizospheres of the treated individuals compared to traditional irrigation. The metagenomic analysis of the rhizospheric communities revealed the survival of the strains in the soil after their addition in any of the chemical treatments without altering the alpha and beta microbial diversity of the rhizospheric communities.

## 1. Introduction

Soil is a limited resource composed of a combination of abiotic elements, such as minerals and organic matter, and biotics, such as the vegetation it hosts and its fauna. It is a layer that has developed slowly over the centuries due to the gradual weathering of surface rocks by various physicochemical agents (water, temperature changes, wind action, among others). Additionally, it has organic elements from the decomposition of plants and animals carried out by microorganisms [1]. Forestry is dedicated to the sustainable use of forests to meet human needs. The potential growth of forests is closely linked to soil quality, so it is essential to maintain their fertility to ensure adequate development of plant species. In Spain, it is estimated that around 20 tonnes of soil are lost per hectare annually due to factors such as urbanization, pollution, and overexploitation [2,3].

Conventional practices for rehabilitating these forest soils include reforestation, erosion management, implementation of vegetation covers, and application of organic amendments. These actions aim to recover the structural quality of the soil, increase its fertilizing capacity, and promote biodiversity [4]. Within the framework of the 2030 Agenda for Sustainable Development, the need for these techniques to be aligned with the objectives of sustainability and efficiency in the use of resources is established. Specifically, the Sustainable Development Goals (SDGs) are proposed, aimed at promoting the circular economy and the use of waste. This is reflected in SDG number 12, focused on responsible production and consumption practices. A good way to align with these objectives for soil recovery could be the use of waste, such as sludge from wastewater treatment plants (WWTPs), to improve plant growth and soil fertility.

Around 1.2 million tonnes of sludge are produced in Spain each year, most of which is landfilled, causing both environmental and economic problems [5], beyond the lost opportunity to generate a value from this waste. This sludge has the potential to be transformed into organic fertilizers and biofertilizers through a recovery process, understood as the process of using and transforming waste into resources, as raw material, to obtain an economic value. These practices contribute to reducing the amount of waste, reducing pollution and promoting the circular economy by promoting the reuse of waste [6], while helping to rehabilitate degraded soils and complete the nutrient cycle. The adoption of technologies for the conversion and reuse of this waste not only decreases the amount of waste, but also promotes sustainable agriculture and more effective management of natural resources [7]. This type of waste; however, must go through a process of adaptation for biotechnological use as fertilizers.

Currently, two of the most environmentally friendly alternatives, as alternatives to traditional agrochemicals, are organic fertilizers and biofertilizers. In the former, organic matter is present in complex forms, so it is not directly accessible to plants. That is why many authors agree on the suitability of adding living microorganisms that, through their metabolism, transform (mineralize) organic matter into bioavailable forms [8,9]. This new formulation is called a biofertilizer. A good example of microorganisms capable of carrying out these processes are the so-called plant growth promoting bacteria (PGPB) [10]. There are various mechanisms by which PGPBs can exercise these actions. On the one hand, there are the direct promotion mechanisms, which are those that occur within the plant cell and it is only its metabolism that is affected. On the other hand, there are indirect mechanisms, which refer to the assimilation of nutrients (through the roots) from the soil, as a result of an increase in their concentration. The beneficial bacteria will produce a signal molecule (known as an elicitor) that will be introduced into the plant’s defense signal transcription pathway [11], thereby altering its secondary metabolism [12]. Two of the principal PGPB used in growth promotion are the genera *Bacillus* and *Pseudomonas.* These bacteria have been shown to be effective both in phosphorus solubilization and nitrogen fixation, as well as in the production of phytohormones, improving the resistance of plants to abiotic stress conditions such as drought, salinity, and high temperatures [13,14]. In addition, these PGPB are well known for producing phytohormones such as 3-indolacetic acid, generating siderophores, solubilizing phosphates, fixing nitrogen, or degrading complex organic substances for an easier uptake by the plants [15,16,17,18].

However, while PGPBs offer numerous benefits, potential risks must be considered in releasing microorganisms to the environment, especially in the context of antibiotic resistance and its transmission. It is well known that soil acts as a reservoir of antibiotic resistance and that horizontal gene transmission occurs in these environments [19,20,21]. This is a threat to global public health. So much so that the World Health Organization (WHO) estimates that antibiotic resistance could cause 10 million deaths per year by 2050 [22]. Resistance mechanisms are shared and transferred between different ecological niches, making them difficult to control. For this reason, a holistic approach that considers human, animal, and environmental health as a whole is required, hence the name of this strategy: “*One Health*”. For all of the above, it must be ensured that PGPB strains with biotechnological applications are free of virulence and antibiotic resistance genes, especially those that are transmissible because they are found in mobile gene elements before their release into the environment.

To study the composition and functionality of their microbiota, advanced techniques are required to solve the problems of classical microbiology, which excludes non-culturable microorganisms that, in essence, constitute the majority. In fact, it is estimated that more than 99% of the microorganisms present in the soil are not cultivable using traditional microbiology methods [23]. Recently, omics techniques, such as metagenomics, have allowed the study of complex environmental samples through the massive sequencing of microbial genomes. This allows us not only to know the state of the soil, but also to study the impact that different fertigation treatments may have on it, as well as its evolution over a given time [24].

For all these reasons, the objectives of this work are to verify the effect of fertigation with valorized WWTP sludge added with the PGPB *Bacillus pretiosus* (C1) and *Pseudomonas agronomica* (C2) on the biometrics and nutritional composition of the holm oak. In the same way, it is intended to study how fertigation with biofertilizer affects the metabolic diversity and antibiotic resistance phenotype of rhizospheric bacterial communities, and to evaluate the survival of the inoculated strains (C1 and C2), as well as its impact on the diversity of the resident soil microbial community.

## 2. Materials and Methods

### 2.1. Bacterial Strains

The strains *Bacillus pretiosus* (C1) and *Pseudomonas agronomica* (C2) were used. These strains had been isolated and characterized by Robas et al. [13]. These strains have biotechnological potential, due to their PGPB activities. Table 1 shows the characterization of these strains. These strains are also free of transmissible antibiotic resistance genes and virulence determinants [13].

### 2.2. Experimental Design: Tested Plants, Substrate, and Chemical and Biological Treatments

The biological tests were carried out on holm oak (acorn) seeds, *Quercus ilex* (L.). supplied by IMIDRA (Madrid Institute for Rural, Agricultural and Food Research and Development). These seeds come from native plants of the Community of Madrid, whose purpose is to repopulate forests to avoid genetic contamination.

They were superficially sterilized with a “30” wash with ethanol (70% *v*/*v*) and washed three times with sterile distilled water. They were then rinsed with plenty of water. Sterilized PVC trays were used for pregermination, which were filled three-quarters with sterilized river sand and saturated with sterile water at field capacity. The seeds were placed on the previously hydrated river sand and covered with filter paper. These trays were kept in darkness at a stable ambient temperature of 20 ± 2 °C for 96 h until a visible radicle of approximately 2 cm ± 0.2 cm emerged. For the preparation of the substrate, forest soil (50%) was mixed with sterilized river sand (50%). The forest soil was obtained from oak groves in the municipality of Navacerrada, in the foothills of the Sierra de Guadarrama (GPS coordinates 40.811938551856144, −3.7920282883570504). The soil was sifted to remove the larger portions and selected to achieve a homogeneous and standard granulometry once mixed with the sterile river sand. Seedbeds of 10 cm × 8 cm with 35 alveoli of 18 cm height were used and placed in forest trays or leachate trays (shared by replicas of the same treatment, to avoid cross-contamination). A total of nine forest trays were used for the different treatments as shown in Table 2. One seed was planted in each alveolus, having 35 seeds per treatment and a total of 315 seeds in the experiment.

In total, three chemical treatments (irrigation matrices) were tested: water (W), WWTP (EDAR) organic waste, and sterilized WWTP organic waste (EDAR_ST). Each of them was tested with three biological treatments: without inoculum (control, C0), supplemented with strain C1 (*Bacillus pretiosus*), and strain C2 (*Pseudomonas agronomica*).

### 2.3. Preparation of Bacterial Suspensions and Biofertilizer

#### 2.3.1. Preparation of Bacterial Suspensions

Starting from pure cultures of the C1 (*Bacillus pretiosus*) and C2 (*Pseudomonas agronomica*) strains on nutritional agar from Condalab^®^ (Madrid, Spain), a bacterial culture was prepared in LB liquid medium. After 48 h of growth, bacterial density was verified using UricultTM (Liofilchem srl, Roseto degli Abruzzi, Italy), which must have been equivalent to 0.5 on the McFarland scale (108 u.f.c. mL^−1^). This process ensured the standardization of the bacterial inoculum for subsequent application in the final volume of the irrigation matrix.

#### 2.3.2. Preparation of the Chemical Irrigation Matrix: Dilution of the EDAR and Sterilization (EDAR_ST)

The WWTP residue (Industrias Cárnicas Villar, S.A., Soria, Spain) demonstrated fertilizing activity experimentally at a dilution of 1/512 (V_EDAR_/VH_2_O). For all the trials in this work, this dilution was used as the basis for the formulation of the biofertilizer (incorporating biological treatment). The physicochemical composition of the WWTP organic waste can be consulted in Table 3, indicating a composition rich in essential nutrients and chemical elements that could have significant effects on biofertilization and soil improvement. To eliminate the microbiota of the WWTP organic waste, an aliquot (EDAR_ST) was sterilized by autoclaving at 121 °C for 20 min at 1 atmosphere of pressure.

#### 2.3.3. Biofertilizer Preparation: Addition of PGPBs to Irrigation Chemical Matrices

The biofertilizer was prepared weekly to avoid non-specific transformations resulting from microbial metabolism (autochthonous microbiota and inoculum). Three liters of each irrigation matrix were prepared in each session and kept refrigerated (4 °C). The volumes of each component of the biofertilizer, to ensure the final microbial density of 0.5 McFarland in the irrigation matrix were: 100 mL of a 0.5 McFarland suspension of each strain (C1 and C2) was added to one liter of the 1/512 dilution of EDAR/EDAR_ST, as appropriate.

#### 2.3.4. Growth Test, Irrigation Protocol, and Plant Growth Conditions

The test was performed under controlled laboratory conditions. It had an overall duration of one year (October 2023–October 2024). The growth phase was carried out in a phytotron (photoperiod of 11 h of light and 13 h of darkness, light intensity of 505 μmoles·m^−2^·s^−1^ (white and yellow light), relative humidity 30 ± 5%). The irrigation regime was designed in order to emulate field conditions and was carried out systematically with the treatments described above (EDAR/EDAR_ST supplemented with strains C1 or C2), and water in the case of the control. The objective was to guarantee the relative humidity of the soil, without reaching saturation or waterlogging, which resulted in a weekly irrigation periodicity with an experimental average volume of 85 mL/alveolus.

### 2.4. Harvesting and Parameter Measurement: Biomass and Biometrics

#### 2.4.1. Harvest

The harvest was carried out after one year of rehearsal. The harvests were destructive and involved the extraction of the aerial and root part from each plant. From the root part, the rhizospheric fraction of the soil (2 g per seedling) was obtained for subsequent analysis.

#### 2.4.2. Biomass Measurement (Dry Weight, g) and Biometrics

For the calculation of the total biomass in terms of dry weight (g), the crop was left to dry at room temperature (22 ± 2 °C) for one week, after which it was weighed on a precision scale. Sampling, which was destructive, was carried out after the one-year of test was completed. They involved the extraction of the aerial and radical part of each plant by manual means, followed by washing with distilled water to remove all traces of substrate. Each plant was treated as a replicate. The length was measured, for the biometric sampling of the root part (cm), length of the aerial part (cm), the number of secondary roots (N°), the number of leaves (N°), the total weight of the plant (g), as well as the weight of the radical part (g) and the aerial part (g).

#### 2.4.3. Nutritional Analysis

Nutritional analysis, carried out 24 h after harvest at the Rock River Labs Spain (Lalín, Pontevedra, Spain), used NIR technology with the NIRS DS2500 equipment (FOSS Analytical, Hillerød, Denmark). The samples were dried, separated (leaves and stems), and grounded with a Cyclotec mill (1 mm sieve) (Foss Iberia, Barcelona, Spain). The NIR spectrum in the range of 400 to 2498 nm was collected and equations from the Rock River Laboratory (Watertown, WI, USA) were applied for analysis. To make the triplicates, the total number of plants per treatment was divided into three groups for processing and analysis. The variables evaluated included water-soluble carbohydrates (WSC), neutral detergent fiber usable at 240 h (uFDN_240), total digestible neutral detergent fiber (TTFDND), soluble protein (Proteína_Sol), crude protein (Proteína_Cruda), non-digestible fraction of neutral-modified detergent fiber (aFDNmo), and total amino acids (AA_Totales).

### 2.5. Analysis of Rhizospheric Bacterial Communities

#### 2.5.1. Extraction of Bacterial Communities

The 35 soils collected per plant were divided into three groups and mixed to form three replicates of 23 g of soil. For the extraction of rhizospheric bacterial communities and free soil, the procedure described in García-Villaraco Velasco et al. [25] was followed and modified. To do this, 2 g of soil per treatment was suspended in 20 mL of sterile saline solution (0.45% NaCl). It was homogenized with an Omni-Mixer shaker (Omni International, Kennesaw, GA, USA) at 16,000 r.p.m. for 2 min. It was then centrifuged at 4500 rpm for 10 min with a Hettich Zentrifugen centrifuge, model Mikro 22R (Hettich Iberia, Gipuzkoa, Spain).

#### 2.5.2. Studying the Metabolic Diversity (Functionality) of the Microbial Community: Biolog Eco^®^ Plates

From the microbial suspension obtained, as described in Section 2.5.1, the 31 wells of the Biolog Eco plates^®^ (Biolog, Hayward, CA, USA) were loaded in triplicates with 135 μL of solution, each containing a different carbon source. The plates were incubated for 168 h at 25 ± 2 °C, and their absorbance was measured at 595 nm every 24 h using the Asys UVM340 plate (Benchmark Scientific Inc., Sayreville, NJ, USA) reading equipment and Micro WinTM V3.5 Software. With the results obtained from the absorbance measurements, the corrected absorbance was calculated by subtracting the target value. Next, the mean corrected absorbance of all wells was calculated as the mean of the 31 wells (Biolog Eco^®^ plates) for each replicate. The AWCD (Average Well Color Development) value [17,18,26] was plotted against the incubation time to obtain microbial community growth curves in the wells of the plates. In these kinetics, the incubation moment at which the growth of the community began the stationary phase was chosen for the subsequent multivariate analyses (144 h). With these AWCDC values, the metabolic diversity of each sample was calculated using the Shannon–Weaver diversity index [27,28] H(m) = −∑ qi log2qi, where qi = n/N, “n” is the corrected absorbance (AWCD) of each well and “N” is the total absorbance of all wells.

#### 2.5.3. Study of Community Antibiotic Resistance: Cenoantibiogram

From each replicate of the soil extract obtained as described in Section 2.5.1, the cenoantibiogram study was carried out following the protocol established by Gonzalez-Reguero et al. [29]. The density of viable microorganisms was found to be approximately 10^8^ cfu mL^−1^ (OD 0.5 McFarland) using UricultTM (Liofilchem srl, Italy). Mueller–Hinton agar (Condalab^®^, Madrid, Spain) was sown on turf and the minimum inhibitory concentration (MIC) was evaluated using ε-test antibiotic strips for the following antibiotics: amoxicillin (AML), amoxicillin-clavulanic acid (AUG), piperacillin (PP), piperacillin-tazobactam (TZP), imipenem (IMI), imipenem-EDTA (IMD), ciprofloxacin (CIP), nalidixic acid (NA), trimethoprim-sulfomethoxazole (TS), cefotaxime (CTX), and cefpirome (CR). The plates were incubated following the manufacturer’s instructions at 28 ± 2 °C (standard growth temperature for environmental bacteria), under aerobic conditions. For the quantification of the minimum inhibitory concentration (MIC), the most restrictive halo was used as a reference. The antibiotics were selected in order to represent those most commonly used in clinical practice.

### 2.6. Study of Taxonomic Composition and Diversity: Metagenomics

The composition and structure of rhizospheric microbial communities were assessed by amplification and sequencing of variable regions V3–V4 of the 16S rRNA gene. Amplification was performed after 25 cycles of PCR. Positive (CM) and negative (NC) controls were used to ensure quality control. The positive control, a mock community, was processed in the same way as the samples. The libraries obtained were sequenced using Illumina Miseq (300 × 2) (San Diego, CA, USA). The analysis was commissioned to the company Microomics^®^ S.L. (Barcelona, Spain), who analyzed the taxonomy of the extracts according to the protocol shown in Figure 1.

### 2.7. Gene Prediction

A functional prediction analysis of genes was performed from sequencing data using PICRUSt2 (Phylogenetic Investigation of Communities by Reconstruction of Unobserved States, version 2). The processing and analysis of the data was carried out in Python 3, using the specific scripts and modules for the execution of PICRUSt2, in order to predict the functional abundance of genes in the analyzed samples.

Before performing the non-parametric Kruskal–Wallis test, the ARiSTa (Adaptive Rank-based Inverse Score Transformation analysis) was applied, to normalize the functional abundances and guarantee the statistical validity of the analysis. After the transformation, a Kruskal–Wallis test was applied to identify differentially abundant functions between the experimental groups.

Finally, a classification model based on Random Forest was implemented to identify the most important functional variables that discriminate between groups.

### 2.8. Statistical Analysis

To know the impact of fertigation treatments on biometric variables (stem and root weight, stem and root length, and number of leaves), an ANOVA was performed, followed by a post hoc analysis (Duncan test). For the study of functional diversity (Biolog Eco plates^®^), the kinetics of the samples subjected to different chemical and biological treatments, in the presence of PGPB inoculums (C1: *Bacillus pretiosus* and C2: *Pseudomonas agronomica*), were first graphed. In order to determine the incubation point of maximum metabolic activity of the community (maximum AWCD value), the slope of each curve was calculated, and the point of maximum slope was selected. At this time, metabolic diversity was calculated using the Shannon–Weaver index. For nutritional analysis, an ANOVA mean comparison test was used to find out if any of the analyzed variables differed depending on the type of chemical and biological treatment. Only for those in which the variations were significant (*p*-value < 0.05), was a post hoc pairwise comparison test (Duncan test) performed in order to study which fertigation treatment justified such differences. Finally, in order to have an integrated view of the grouping trends of nutritional variables according to chemical and biological treatments, a representation of principal component analysis (PCA) and the respective load factors were carried out. To study the effect of the different treatments on the antibiotic resistance profile of the microbial community (cenoantibiogram), an ACP was used as an exploratory technique of trends and to discriminate differences between treatments, accompanied by a graph of the load factors. For all analyses, the statistical program SPSS v.29.0 (IBM Corp, Armonk, NY, USA) was used.

## 3. Results

### 3.1. Plant Biometrics Analysis

The biometric analysis was performed with n = 35 for each treatment. Figure 2 shows how organic fertilizer treatments (EDAR, EDAR_ST) improved the development of biometric variables significantly compared to water controls (WC0). Figure 2A shows an increase in the variables of dry weight of the stem and dry weight of the leaves due to the addition of EDAR residue and EDAR_ST. Likewise, at the intratreatment level, we can observe how the addition of PGPB strains (C1 and C2) significantly increased the value of these biometric variables with respect to their controls. In a global analysis, we can highlight the action of the combination of EDARC2 and EDAR_ST with both strains in biometric weight improvement.

Figure 2B shows, in a similar way to Figure 2A, that the variables of stem length and root length are significantly increased when the residue under study is added, both in its raw (EDAR) and sterilized (EDAR_ST) forms, compared to traditional irrigation with water. Likewise, the addition of C1 and C2 strains significantly increased the root and stem length compared to their controls.

In the sheets (Figure 2C), we find the same trend as in the previous cases (Figure 2A, B). A significant increase in the number of leaves in individuals was observed according to chemical treatments (EDAR and EDAR_ST) compared to traditional irrigation. Similarly, an increase in the number of leaves in biological treatments was observed compared to their controls. In this case, those treated with EDAR_STC1 and WWTP with both bacterial inoculums stand out.

### 3.2. Analysis of Nutritional Quality in Stem and Leaves

All the nutritional parameters that showed statistically significant differences between treatments (*p*-value < 0.05) in the ANOVA were studied in depth. For those variables that were most influenced by fertigation treatment, a post hoc study was carried out using the Duncan test.

In Figure 3 of the protein parameters, the same trend was generally observed as in the biometric variables. Figure 3A shows an increase in the protein variables in the leaf in the EDAR and EDAR_ST treatments compared to the water control. In the intratreatment analysis, it was observed that the addition of the strains has a favorable effect on the protein content compared to the controls without inoculation. The same results were observed in Figure 3B of the amino acid content. These variables (Figure 3A,B) showed a significant improvement in protein content in the EDAR_STC1 and EDAR_STC2 treatments.

This same trend can be observed in Figure 4 of the protein parameters in stem. Also noteworthy is a significant improvement in protein content in EDAR_STC1 and EDAR_STC2 treatments. The rest of the measured parameters (Appendix A) show similar trends with small variations. It stands out in all of them how the addition of bacterial inoculum significantly increases the nutritional parameters measured with respect to the controls.

It should be noted that, in the stem (Figure 4), the relative content of lignin and starch follows the same trend as the biometric parameters. Not only do plants have a larger size when they are irrigated with chemical and biological treatments, but they also have a higher content of lignin and starch per gram of plant. Figure 4A–C show how chemical treatments (EDAR and EDAR_ST) generally and significantly improve protein, amino acid and carbohydrate content compared to irrigation with water. As an intratreatment, these variables are increased by the presence of C1 and C2 compared to controls without bacterial inoculum (C0).

In order to analyze all the nutritional parameters in a more integrated way, Figure 5 below represents the two-dimensional distribution of the variables, for the leaf (Figure 5A,B) and for the leaf (Figure 5C,D), depending on the type of fertigation treatment used.

In Figure 5A, the PCA projection of the two variables that explain the model (cumulative variance of 68.67%) for the stem nutritional variables can be seen. As can be seen, those plants that did not receive inoculum, regardless of chemical treatment, are grouped in the central area of the graph. In this way, there is a clear segregation of the data according to the type of biological treatment. Likewise, those treated with the C1 and C2 strains are segregated toward the extremes. Figure 5B shows the chemical variables that are heterogeneously grouped in the graph.

The PCA in reference to the analysis of the nutritional parameters of the leaf can be seen in Figure 5C,D. Figure 5C shows the PCA with the projection of the two variables that explain the model (cumulative variance of 63.63%). We can see how the individuals are distributed in this analysis in the same way as in Figure 5A, with the control treatments (C0) grouped in the central area of the graph and the treatments with C2 and C1 segregated at both poles of the *x*-axis. Figure 5B shows the chemical variables, which are grouped heterogeneously in the graph, with a greater tendency of distribution towards the positive axis of the *x*-axis. This distribution is due to the significant differences found in the ANOVA figures when we compare the weight of each of the variables (Figure 5B) with the distribution of the individuals (Figure 5A), finding that treatments with C1 (*Bacillus pretiosus*) had a higher content in the nutritional variables analyzed compared to the rest of the treatments.

### 3.3. Studying the Metabolic Diversity (Functionality) of the Microbial Community: Biolog Eco^®^ Plates

The metabolic diversity of each sample was calculated (Table 4) using the Shannon–Weaver diversity index, and differences were analyzed at 144 h of incubation, when the community had entered the stationary phase of growth. The kinetics of the 168 h of measurements can be seen in Appendix A.

The results of the calculation of the Shannon index show that there are no statistically significant differences between the different chemical and biological treatments. Functional diversity values are greater than zero, suggesting that they are metabolically active communities with a relatively equal distribution of functional activity. The addition of the C1 and C2 strains, on either of the chemical matrices, does not modify the functional diversity significantly.

### 3.4. Study of Community Antibiotic Resistance: Cenoantibiogram

Figure 6A represents the PCA with the projection of the two variables that best explain the model (cumulative variance of 85.27%). A clear segregation of data was observed according to the type of biological treatment, which translates into phenotypes of resistance to the different antibiotics tested. In this way, those plants that did not receive inoculum, regardless of chemical treatment, are grouped in the right half of the graph. Likewise, those treated with strains C1 and C2 are segregated towards the left half. Among these, there is also another trend towards segregation between biological treatments C1 (in green) and C2 (in red). Figure 6B shows the load factors. All the antibiotics tested seem to be grouped in the right half of the graph, suggesting that they are the ones that condition the segregation towards higher values for the treatments without added strains (C0).

### 3.5. Study of Taxonomic Composition and Diversity: Metagenomics

#### 3.5.1. Quality Analysis

Table 5 and Table 6 represent the quality indices of the sequences, for analysis prior to the bioinformatics study. These results indicate uniform and sufficient coverage between samples, with an average frequency of (32,328), ensuring adequate depth for further analysis. The values of the first and third quartile reflect a consistent distribution of data, while the median and mean suggest that there are no significant biases in the coverage of the sequences. With this data quality, it is possible to proceed with confidence to the bioinformatic analysis.

#### 3.5.2. Beta Diversity

Beta diversity was analyzed using an ACP with the Bray–Curtis metric (Figure 7) to evaluate dissimilarities in taxonomic composition between rhizospheric communities subjected to different chemical and biological treatments. Figure 7 shows a segregation of data (related to taxonomic diversity) according to biological treatment. The variability of samples treated with C1 and C2 is more captured by the axis and suggesting that there may be patterns of variability common in the communities, or at least differences from those that have not been treated with PGPB (C0).

#### 3.5.3. Relative Abundances at the Genus Level

In order to understand the impact that the addition of the strains (C1 and C2) has on the taxonomic composition of the soil microbial community, a metagenomic analysis was performed by sequencing 16S rRNA amplicons. The results at the genus level are represented in Figure 8.

The results show relative increases in the genus *Pseudomonas* in soils fertigated with the C2 strain, regardless of the chemical treatment that conveys it. However, the increases in the soils fertigated with the C1 strain show a notable increase in the amount of *Bacillus.* Based on these results, the survival of the added strains is evidenced. However, no displacement of the rest of the taxa of the soil microbial community that hosts them was observed.

### 3.6. Gene Prediction

A functional analysis based on KEGG annotations was carried out to characterize the metabolic potential of the microbial communities present in the different treatments: chemical fertilizer (Chemical_Fertilizer), ORGAON^®^PK biofertilizer (ORGAON_PK), its sterilized version (ORGAON_PK_ST), and water treatment (Water). Functional profiles were grouped into major metabolic pathways and visualized using relative abundance plots (Figure 9).

Overall, the microbiomes of all treatments presented a similar functional profile, characterized by a high representation of pathways associated with amino acid metabolism, carbohydrate metabolism, energy metabolism, and cofactor and vitamin metabolism. To a lesser extent, functions related to the biosynthesis of secondary metabolites, lipid metabolism, nucleotide metabolism, and degradation of xenobiotic compounds were observed.

A differential analysis of KEGG functions among microbial communities was performed using the nonparametric Kruskal–Wallis test. This analysis revealed thousands of statistically significant functions after correction for multiple tests (FDR < 0.05). In order to reduce the dimensionality of the dataset and prioritize those functions with greater discriminative capacity between treatments, a Random Forest classification model was applied to the KO functions predicted by PICRUSt2. This approach made it possible to identify the most relevant functional genes for differentiation between experimental groups.

From the functions noted by COG categories, a relatively homogeneous functional distribution was observed among the treatments. The most abundant functions included those related to amino acid and carbohydrate transport and metabolism, energy production and conversion, cell envelope biogenesis, and signal transduction. The categories “unknown function” and “general prediction of function” were also highlighted, reflecting the presence of genes with incomplete or poorly characterized annotations.

The least represented categories were those related to RNA processing and modification, post-translational modification, intracellular trafficking, defense mechanisms, and biosynthesis of secondary metabolites. No drastic differences were observed between treatments, although certain functional profiles showed subtle variations in the relative abundance of pathways associated with regulatory and structural processes (such as DNA replication and repair or ribosomal translation).

A Random Forest classification analysis was applied using the KO functions predicted by PICRUSt2, with the aim of identifying the most relevant functional genes for discrimination between treatments (Figure 10). The results show that the model reached an overall classification error (OOB error) of 25.9% and allowed the identification of a set of KO functions with a high contribution to the accuracy of the model, measured as “Mean Decrease Accuracy”.

Each treatment had specific highly classified functions (Table 7). For example, in the WC0 treatment, KO K12645 stood out, corresponding to epi-isozizaene 5-monooxygenase/beta-farnesene synthase (EC:1.14.15.39/4.2.3.47), enzymes associated with terpene biosynthesis pathways. In WC1 and WC2, functions linked to membrane proteins and ubiquinone biosynthesis (K11929 and K18587, respectively) were identified.

In the EDARC0 treatment, KOs related to sporulation (K06390) and antibiotic resistance (K10012) were identified. In EDARC1, KO K02777, corresponding to a component of the PTS sugar transport system, was the most relevant. On the other hand, in EDARSTC1, functions associated with anaerobic sulfite reduction (K00385), bacitracin transport (K19309), and regulation of phosphotransferase signaling (K06368) were detected. These functions suggest differential functional adaptations related to the presence of the WWTP residue and the inoculated strain.

Finally, to provide a more detailed representation of the microbial functionality, KEGG modules and complete KEGG pathways were analyzed (Appendix A). These complementary graphs allow us to visualize the presence of complete routes and specific modules that could be relevant in subsequent studies.

## 4. Discussion

*Quercus ilex*, known as holm oak, is crucial for biodiversity and ecological stability in Spain’s Mediterranean ecosystems. However, the loss of biomass and the degradation of their habitats demand effective methods of conservation and restoration [22,23,24]. The present study is postulated as a prospective to develop innovative approaches to recover degraded soils where *Quercus ilex* is a plant species of the phytosociological series, thereby promoting sustainable practices to ensure the preservation of this valuable natural resource in Spain**.**

PGPBs are capable of promoting plant growth by various mechanisms, both direct and indirect. The direct mechanisms are those where these microorganisms stimulate plant development through the production of growth regulators (auxins, cytokinins, gibberellins, abscisic acid), biological nitrogen fixation, solubilization, and phosphate mineralization [9]. In the present work, and in line with many other authors, two genera that are widely described for their PGPB capabilities, ecological versatility, metabolic plasticity, and safety were tested: *Pseudomonas* sp. and *Bacillus* sp. [30,31].

Likewise, the use of WWTP waste fulfills several functions within the framework of this work. On the one hand, it acts as a vehicle matrix for the bacterial strains under study; on the other hand, it provides a complex organic substrate that the strains can use to transform from organic to inorganic forms, due to the mineralizing action of microorganisms, thereby facilitating their absorption by plants [27,28,32]. Other authors have worked with similar residues in the production of fertilizers and biofertilizers [29,33,34], in order to reintroduce these wastes into the production chain.

The bacterial genera used in the present study have been extensively studied and used for the promotion of plant growth [30,35]. As can be seen in the results obtained, there is a significant increase in the biometric variables measured when both PGPBs are inoculated in isolation and conveyed in the biofertilizer. Other authors have seen similar effects with the use of complex matrices and the use of PGPB bacterial strains in pine and oak reforestation projects [31,33,36]. Similarly, the increase in carbohydrate and lignin content shows a higher nutritional health in *Q. ilex* compared to plants under traditional irrigation. This fact postulates that treatment with biofertilizer not only promotes larger plant growth but also enhances robustness.

The chemical treatment used, in any of the biological treatments tested (EDAR and EDAR_ST), induced an improvement in protein content, compared to the control irrigated with water, a fact already described by other authors [37]. In the treatments to which both PGPB strains were added, a statistically significant increase was observed with respect to their matrix controls without inoculum, both for soluble protein and crude protein. Specifically, the soluble protein is the one that is able to adopt a globular confirmation in water, this is due to free radicals (-R) of amino acids that, upon ionization, establish weak bonds (hydrogen bonds) with water molecules. This property is what makes it possible to hydrate the tissues of living beings [38]. This fact helps to explain the better metabolic development of the plant at its most mature stages and may explain the increase in dry weight observed. The authors attributed this to an improvement in de novo synthesis in the leaves, favored by the direct effects of PGPBs on the plant.

As for amino acids, it can be observed that there is a slight improvement whenever a different treatment is used as compared to water, whether it is the EDAR or the EDAR_ST, and this improvement appeared to be independent of the strains. The clover plant is a legume that is very rich in protein and therefore in amino acids [39].

The diversity of microorganisms, both in their composition and in their metabolic activity, is essential for the maintenance of the health and quality of ecosystems, since a wide variety of microorganisms are involved in important functions and soil transformations [39,40,41,42]. In addition, the diversity of the strains present can be key to the ability to suppress diseases of plants transmissible by this route, so the evaluation of how this diversity is affected in arable soils is essential [43]. From the point of view of the composition of microbial communities, it is known that the abrupt incorporation of an organism can reduce the biological diversity of a system [44,45,46]. However, this is not an universal norm. In the field of microbial ecology, the introduction of a new taxon can lead to an increase in this diversity, as long as there is no displacement of the rest of the taxa in the community [47]. The most frequently assessed aspects of diversity are species richness (or number) and the proportional distribution of the number of individuals of each species. These measurements are a way of describing ecological communities, in terms of dominance or equity, as another component of diversity. In the field of microbial ecology, many indices for measuring biodiversity have been proposed and are widely available [46]. In view of the results obtained in this work, through the use of Biolog Eco^®^ plates, it can be stated that the exogenous contribution of both bacterial strains, C1 and C2, does not modify the metabolic diversity in any of the chemical and biological treatments, with respect to their corresponding controls. This shows that these strains do not displace native microbial communities but manage to coexist with them, respecting their metabolic relationships and the natural balance of the soil. This has already been demonstrated in other studies. For example, the inoculation of a consortium of *Pseudomonas* sp. and *Azotobacter* sp. in wheat [16], and similarly, in oilseed rape, a consortium with *Pseudomonas* sp. increased the relative abundance of some beneficial groups without reducing the diversity of the native community [28]. The method used to estimate this diversity remained at similar values between treatments, suggesting that the introduction of treatments does not alter the harmony of the microbiome. This finding is relevant, as it assures us of the preservation of the metabolic relationships already established and the ecological balance of the soil.

For the biotechnological use of these strains to be safe and beneficial to the environment, it is necessary to ensure that the strains used are harmless. One of the elements that argue in favor of its safety is the absence of mechanisms of resistance to transmissible antibiotics [48]. In this way, the innocuousness of these strains with respect to the variations in metabolic diversity must be analyzed from other approaches. A novel approach is the study of antibiotic resistance in the soil community [21,31], since stability in diversity itself implies a positive consequence if it entails a biological threat [38,49].

Soil microorganisms produce a wide range of secondary metabolites that are fundamentally involved in communication, competition phenomena, and adaptation to environmental changes [43,50]. To gain a competitive advantage, some microorganisms are able to produce antimicrobial compounds to inhibit the growth of their competitors [51]. Additionally, antimicrobial-producing microorganisms may possess self-protection and resistance mechanisms, which allow them to effectively defend themselves against the action of antimicrobial compounds [52]. Thus, the addition of this type of bacteria can induce changes in the behavior of the soil microbial communities that host them, for example, by modifying the antibiotic resistance profiles. One way to quantify this would be through the calculation of the community MIC by means of the cenoantibiogram technique [21]. The low MIC values of the strains used postulate them as good candidates for environmental use, as they reduce the potential risk to human health. This approach to assess the state of soil health has already been used previously in similar trials by other authors [15,21,27,31,48,53,54,55]. In the present work, the global analysis of the MICs for the antibiotics tested in the different chemical and biological treatments showed a separation of groups according to biological treatment. Thus, it can be observed how the MIC profiles of the populations of soils treated with C1 (*B. pretiosus*) and C2 (*P. agronomica*) decreased compared to those that were not treated with the bacterial inoculum. This fact shows the influence of the added strains on treated soils, resulting in a greater impact on the MICs of biological treatments than chemical treatments. These effects have already been observed in other studies in which these same strains were used [15,31], decreasing the MIC of the rhizospheric community that hosts them and potentially reducing the horizontal transfer of these resistances [56,57,58].

Microbial diversity is an indispensable piece of knowledge for understanding the functioning of ecosystems, as microorganisms are a fundamental component in biogeochemical processes [59]. However, this component was overlooked for years in the scientific literature, due in part to the lack of tools to analyze and model the role of microorganisms in ecosystems [60,61,62]. In the last two decades, a wide variety of molecular tools has emerged for the study of the composition of complete microbial communities [60]. These technological advances have opened the door to an unprecedented growth of ecological studies on microorganisms in their natural environments [63].

Metagenomics assays have been a fundamental tool for assessing the impact of different treatments, both chemical and biological, on soils [56]. The taxonomic profile of a microbial community can be obtained by sequencing amplicons of the 16S ribosomal RNA gene. It has been shown that the sequencing approach of this amplicon produces results which are quantitatively and qualitatively different from other methods [56]. This technique is based on the principle of finger printing, allowing us to have qualitative and quantitative evidence of the taxonomy of the strains present in the studied soil [64]. It is important to demonstrate that the changes produced in the studied plants are a consequence of the addition of the PGPBs that are studied, so metagenomics plays a key role in the context of the present study. In addition, it is interesting to check the interaction of microorganisms with each other and what is the mechanism of action or the pathways involved in improving the properties [65,66]. In the case of our results, there is no doubt that the addition of the C1 and C2 strains represents an increase in the population of the species *Bacillus* sp. and *Pseudomonas* sp., compared to the controls (in which the natural presence of these strains and their quantity is seen). Therefore, it can be stated with genetic evidence that, firstly, the added strains remain in the soil, and secondly, that they have adapted to the ecosystem without producing alterations to the rest of the microbial communities. It is important to note that, despite having introduced two exogenous species, the taxonomic diversity is not modified, being very positive since the strains have not displaced the resident microbiota, maintaining the microbial architecture of the soil. Once the strains have exerted their effect and are no longer added, the soil will return to its natural state, as previously described by [67]. Various metagenomic studies on the effects of PGPBs on plants have focused on identifying why these bacteria lead to such important changes, despite having a small inoculum in many cases [68,69]. A study by Kuramae et al. [70] concludes that the significant differences in treatments with the added strains are due to the fact that the rhizome and endophytic microbial communities both play equally important roles in the complicated plant–microbe interactions.

The functional outcomes predicted by PICRUSt2 reveal a conserved functional structure between treatments, dominated by central metabolic functions. This pattern suggests that, despite differences in microbiome composition induced by the WWTP residue and the inoculated strains, bacterial communities maintain essential trophic functions. This functional stability could be related to the functional redundancy typical of soil microbiomes, where different microbial species can fulfill similar functions.

The differences observed in the relative representation of pathways related to energy metabolism, biosynthesis of secondary metabolites, or signal transduction, could reflect adaptive responses to the compounds present in the WWTP residue. In particular, the modulation of pathways associated with antibiotic resistance, transport, and reduction in redox compounds could be linked to selective pressures derived from the organic matrix of the residue.

The use of specific bacterial strains, such as *Bacillus pretiosus* or *Pseudomonas agronomica*, may have induced distinctive functions in some treatments, including enrichment of sporulation-linked functions, active transport, and post-translational modification of proteins. These functions may be indicative of a functional response of native communities to microbial competition or synergy.

The combination of classical statistical analysis (Kruskal–Wallis) with classification approaches such as Random Forest allowed not only the identification of differentially abundant functions, but also the prioritization of those with greater discriminative capacity between treatments. This integrative strategy favors a more robust interpretation and is oriented towards the selection of functional biomarkers relevant for future applications in biostimulation or bioremediation.

Although the data comes from a functional prediction based on the 16S rRNA gene, the observed patterns are consistent with metagenomic studies in soils treated with organic residues. Experimental validation of these functions by metatranscriptomics or metabolomics would be a logical step to confirm actual functional activity. Likewise, integrating these data with soil indicators and plant performance parameters will allow for a better contextualization of the functional impact of the treatment on the agroecosystem.

Some limitations can be addressed to the results obtained. It would be necessary to conduct more research with different soils and more individuals per treatment to confirm the present results. The MIC reduction should also be addressed in different conditions and checked if it is kept in time. In the same way, field tests would be necessary to confirm the effects of the bacterial and biofertilizer under natural field conditions.

## 5. Conclusions

The addition of the PGPB *Bacillus pretiosus* (C1) and *Pseudomonas agronomica* (C2) does not induce a depression in metabolic or taxonomic diversity in the soils in which they are added to. This fact shows the rapid adaptation of the strains to the inoculated soils without producing a negative impact on them.

The segregated projection in the PCA of the results, based on their composition as a fertilizer versus biofertilizer, shows the influence of added PGPBs on soils, as well as the beneficial effects related to the reduction in MIC and the increase in functional diversity.

Metagenomic analyses show the survival of the strains in the soil after their addition in any of the chemical treatments. This fact may explain both the response of the reduction in the minimum inhibitory concentration in the tested soils and the increase in functional diversity. In addition, relative increases in the genus *Pseudomonas* are evident in those soils fertigated with the C2 strain, regardless of the chemical treatment that conveys it.

## Figures and Tables

**Figure 1 life-15-01490-f001:**
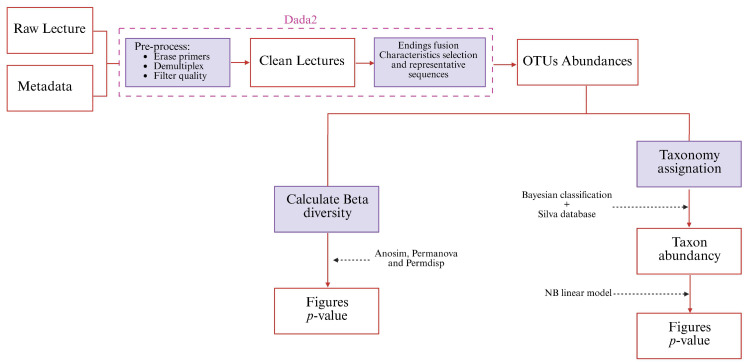
Sequence processing and analysis sequence.

**Figure 2 life-15-01490-f002:**
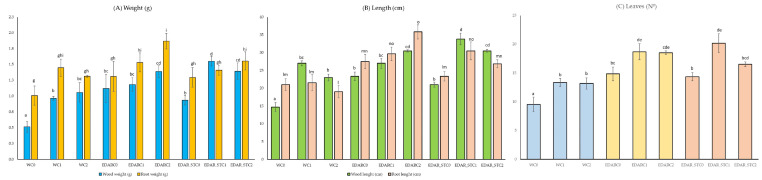
Biometric results. The different letters indicate statistically significant differences (*p*-value < 0.05). (**A**) Dry weight of stem (a–d) and root (g–i); (**B**) Stem (a–d) and root (l–o) length; and (**C**) Number of leaves (a–e). C1: *B. pretiosus*; C2: *P. agronomics*; and W: water).

**Figure 3 life-15-01490-f003:**
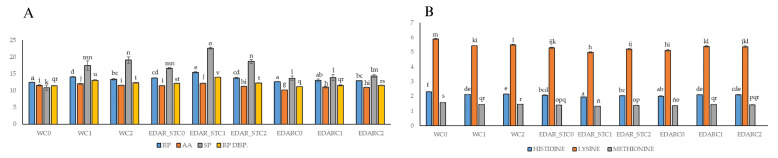
Acorn-fed holm oak leaf statistics. Mean nutritional variables (n = 3) related to the quality of proteins. (**A**) Raw Protein (RP) letters a–e; Amino Acids (AA) letters g–j; Soluble Protein (SP) letters k–o; Raw Protein Available (RP DISP) letters q–v. (**B**) Histidine letters a–f; Lysine letters h–m; Methionine letters ñ–s. Bars with identical letters indicate that the average values are not significantly different (*p*-value < 0.05). C1: *Bacillus pretiosus*; C2: *Pseudomonas agronomica*; W: watering with water.

**Figure 4 life-15-01490-f004:**
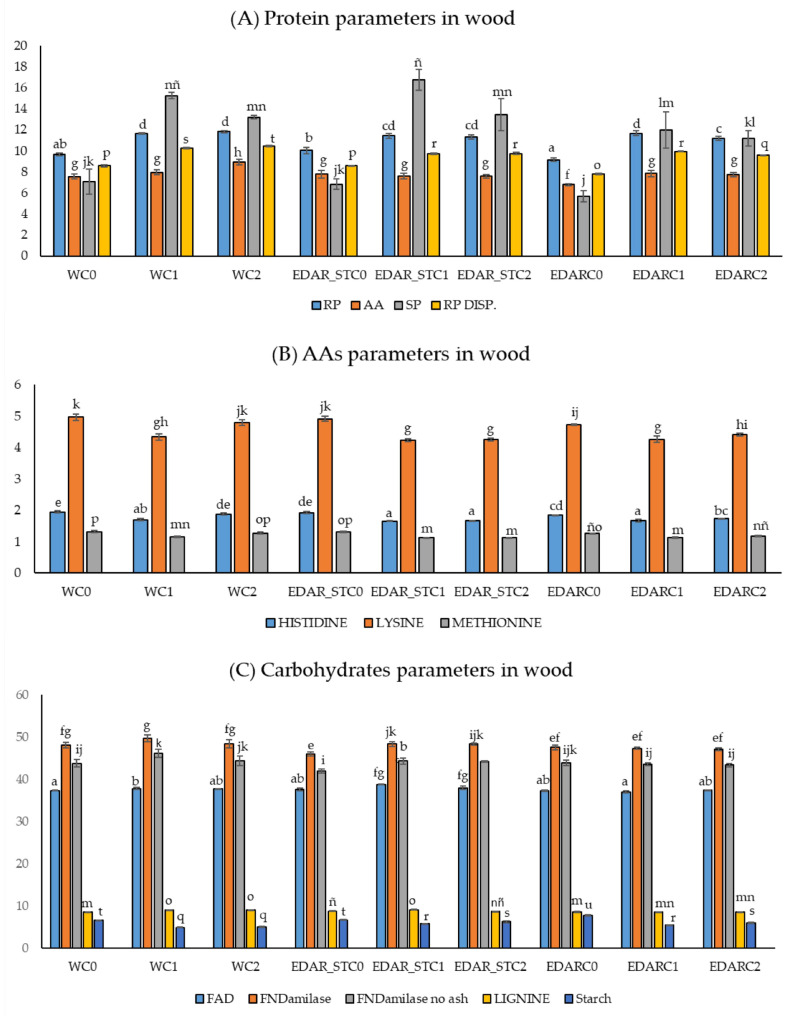
Acorn-fed holm oak wood statistics. Mean nutritional variables (n = 3) related to the quality of protein, carbohydrates, minerals, fatty acids, and fibers. Bars with identical letters indicate that the average values are not significantly different (*p*-value < 0.05). C1: *Bacillus pretiosus*.; C2: *Pseudomonas agronomica*; W: watering with water.

**Figure 5 life-15-01490-f005:**
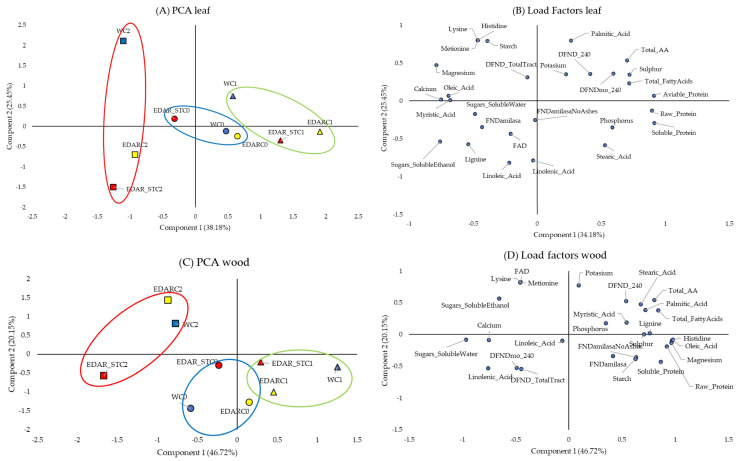
Representation of the nutritional variables of the stem in a two-dimensional plane. (**A**,**C**) ACP which represents the distribution and variation trends of chemical and biological irrigation treatments, according to the two components (variables: nutritional) of the leaves and the stem in a two-dimensional plane. Surrounded in blue: grouping of the treatment controls. Surrounded in green: grouping of the treatments added with strain C1: *Bacillus pretiosus*. Surrounded in red: grouping of the treatments added with strain C2: *Pseudomonas agronomica*. (**B**,**D**) Leaf and stem load factors, respectively.

**Figure 6 life-15-01490-f006:**
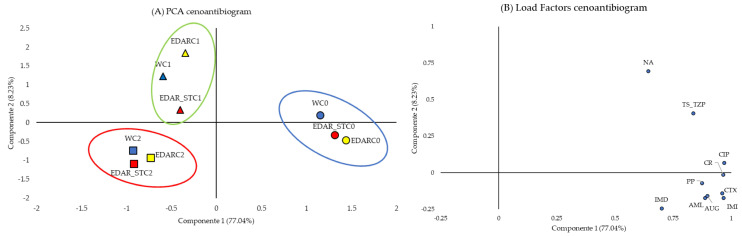
(**A**) PCA: Projection on the two-dimensional plane of the antibiotics tested according to the type of fertigation. (**B**) Load factors. C1: *Bacillus pretiosus*; C2: *Pseudomonas agronomica*; W: watering with water; Antibiotics: amoxicillin (AML), amoxicillin-clavulanic acid (AUG), piperacillin (PP), piperacillin-tazobactam (TZP), imipenem (IMI), imipenem-EDTA (IMD), ciprofloxacin (CIP), nalidixic acid (NA), trimethoprim-sulfomethoxazole (TS), cefotaxime (CTX), and cefpirome (CR).

**Figure 7 life-15-01490-f007:**
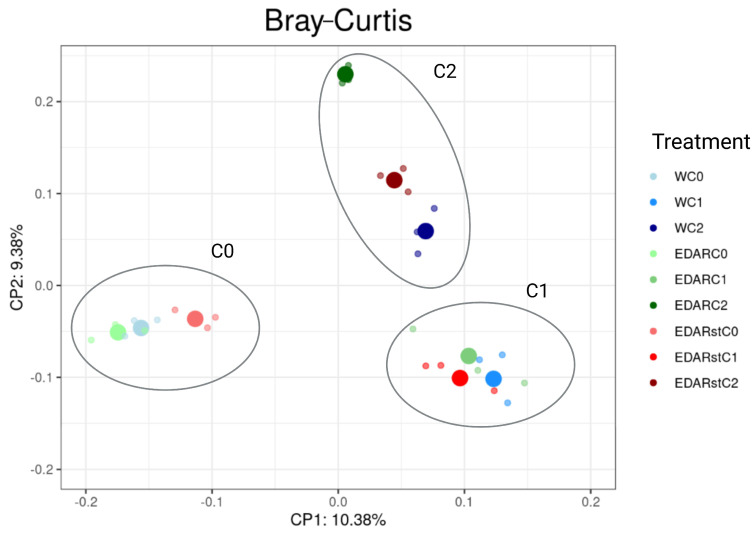
ACP representing the two-dimensional distribution and variation trends of taxonomic diversity (Bray–Curtis) in the samples, depending on the chemical and biological treatment, which best explain the model. C0: control without inoculum; C1: *Bacillus pretiosus*; C2: *Pseudomonas agronomica*; W: watering with water. Surrounded in blue: grouping of chemical treatments irrigated with fertilizer (EDAR). Surrounded in green: grouping of chemical treatments irrigated with sterilized fertilizer (EDAR_ST). Surrounded in gray: grouping trends.

**Figure 8 life-15-01490-f008:**
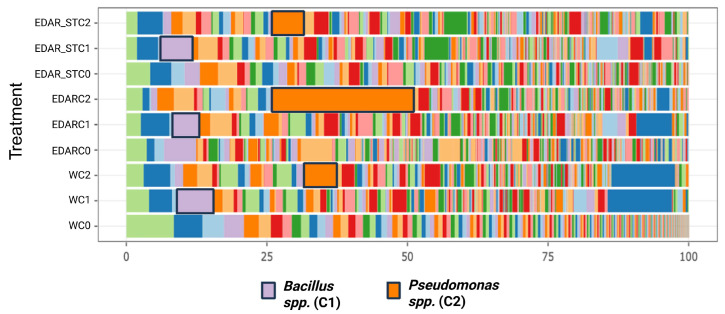
Relative abundances of taxonomic composition at the genus level of rhizospheric samples of *Q. suber* under the different chemical and biological treatments. C0: control without inoculum; C1: *Bacillus pretiosus;* C2: *Pseudomonas agronomica*; W: watering with water.

**Figure 9 life-15-01490-f009:**
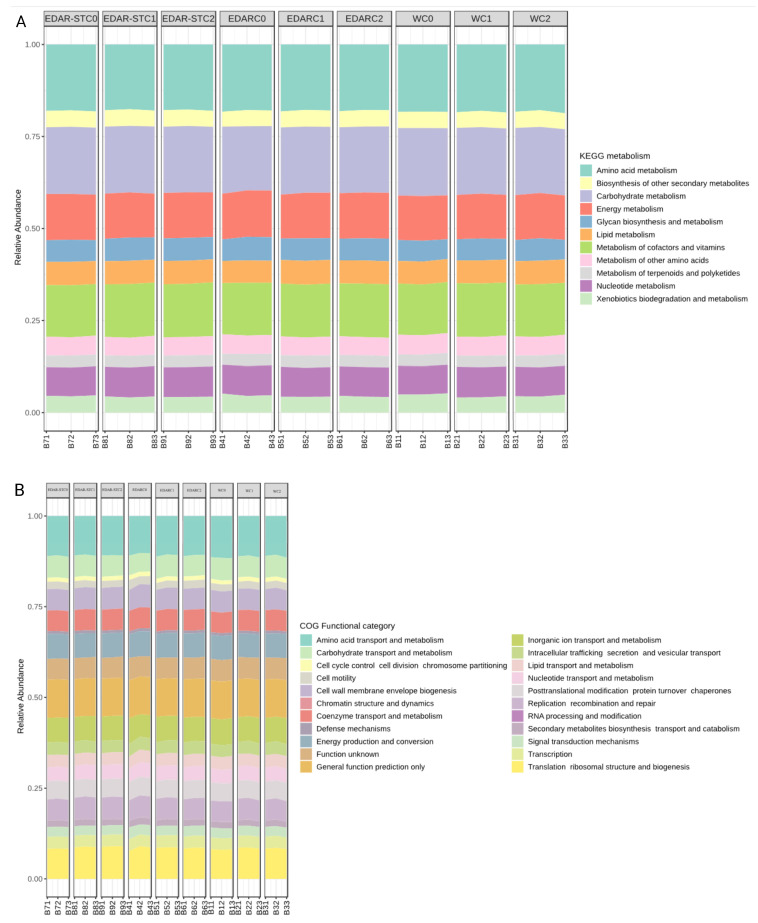
(**A**) General functional profile of KEGG metabolic pathways predicted by PICRUSt2 in microbiomes associated with water and biofertilizer treatments (sterilized and non-sterilized). The main routes correspond to level 2 categories of the KEGG system. (**B**) Functional distribution of metabolic pathways grouped according to KEGG classification at the category level.

**Figure 10 life-15-01490-f010:**
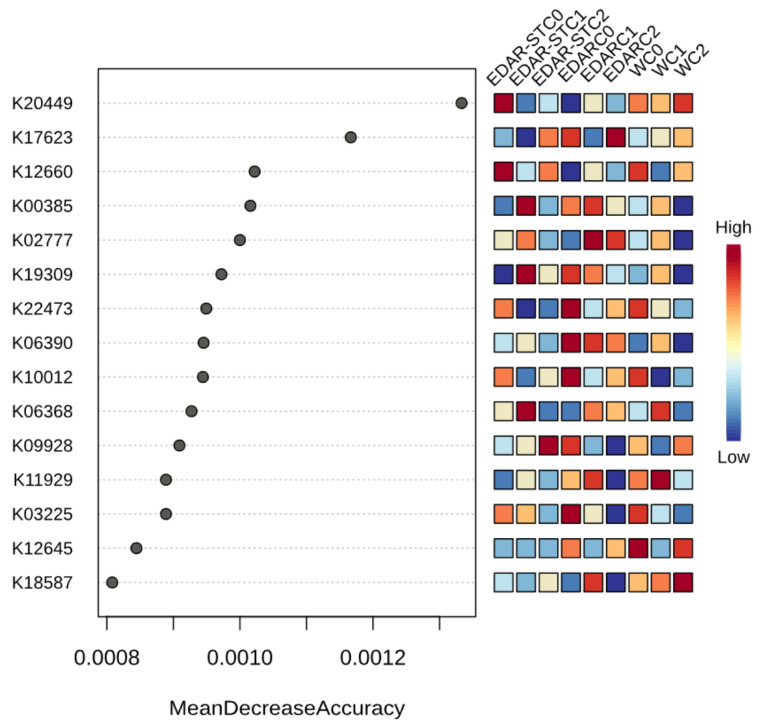
KEGG functions of greatest importance in the Random Forest classification model for discrimination between treatments. The functions represented correspond to the KOs with the highest “Mean Decrease Accuracy” values in the model.

**Table 1 life-15-01490-t001:** The origin, identification, and characteristics of the strains tested.

ID (WGS)	Origin of the Insulation	IAA (μg/mL)	SID (p/a)	ACCd (p/a)
***Bacillus pretiosus*** **(C1)**	Bulk soil	5.61 ± 0.03	+	+
***Pseudomonas agronomica*** **(C2)**	*Medicago sativa*	5.85 ± 0.09	+	+

IAA: production of 3-indoleacetic acid; SID: production of siderophores; ACCd: production of ACC deaminase; p/a: for presence/absence. WGS: *Whole genome sequence*. +: means presence in this context.

**Table 2 life-15-01490-t002:** Experimental design: chemical and biological treatments.

		Chemical Treatment		
		Water	EDAR	EDAR_ST
**Biological Treatment**	**C0**	WC0	EDARC0	EDAR_STC0
	**C1**	WC1	EDARC1	EDAR_STC1
	**C2**	WC2	EDARC2	EDAR_STC2

C0: control without inoculum.; C1: *Bacillus pretiosus*; C2: *Pseudomonas agronomica*; W: watering with water. EDAR: Wastewater treatment plan. Treatments include chemical treatment (irrigation matrix) and biological treatment (C0, C1, C2).

**Table 3 life-15-01490-t003:** Physicochemical composition of the WWTP waste (analysis carried out by LabAqua; tests covered by accreditation ENAC n°109/LE 28).

	Parameters	Methods	Results	Units
**Physicochemical Characteristics**	Conductivity at 20 °C	A-F-PE-0015 Electrometry	1454	µS/cm
	Conductivity at 25 °C	A-F-PE-0015 Electrometry	1612	µS/cm
	Biochemical Oxygen Demand (BOD5)	A-F-PE-0002 Gauge method	3200	mgO_2_/L
	Chemical Oxygen Demand	A-F-PE-0003 Digestion–Colorimetry	6720	mgO_2_/L
	Chemical demand for decanted oxygen	A-F-PE-0003 Digestion–Colorimetry	4220	mg/L
	Nitrates	A-F-PE-0010 Digestion	<0.05	mg/L
	Kjeldhal Nitrogen	A-F-PE-0007 Kjeldhal	296.7	mg/L
	pH	A-F-PE-0010 Electrometry	6.5	U.pH.
	Suspended solids	A-F-PE-0006 Gravimetry	3228	mg/L
	Toxicity	PIT-F/0012 Bioluminescence assay with *Vibrio fisheri*	14	U.T.
**Majority Cations**	Potassium	A-D-PE-0025-ICP-OES	60.2	mg/L
**Anions**	Nitrates	A-BV-PE-0001 HPLC–Conductivity	<2.5	mg/L
	Orthophosphates	Ca-R-PE-0011 Spectrometry	74.32	mg PO_4_/L
	Sulfates	A-BV-PE-0001 HPLC–Conductivity	89.0	mg/L
	Sulfites	A-F-PE-0040 Volumetry	4.5	mg/L

**Table 4 life-15-01490-t004:** Rhizospheric community metabolic diversity at 144 h of growth.

	WC0	WC1	WC2	EDARC0	EDARC1	EDARC2	EDAR_STC0	EDAR_STC1	EDAR_STC2
H’144 h	4.59 ± 0.08	4.47 ± 0.06	4.79 ± 0.05	4.49 ± 0.11	4.75 ± 0.06	4.23 ± 0.07	4.57 ± 0.08	4.38 ± 0.06	4.86 ± 0.04

Representation of mean values per treatment (n = 3). C0: control without inoculum; C1: *Bacillus pretiosus*; C2: *Pseudomonas agronomica*; W: watering with water.

**Table 5 life-15-01490-t005:** General metrics.

Metric	Value
No. of samples	27
No. of phylotypes	10,699
Total frequency	872.77

**Table 6 life-15-01490-t006:** Frequency by samples.

Type	Frequency
Minimum frequency	20,614
First quartile	28,475
Median frequency	31,939

**Table 7 life-15-01490-t007:** Functions predicted to be of greatest importance in the Random Forest model according to the treatment applied. The KO genes identified as most relevant for the classification of treatments by the Random Forest algorithm are shown, along with their functional function annotated in KEGG.

Treatment	KO	Predicted Function (KEGG)
WC0	K12645	epi-isozizaene 5-monooxygenase/beta-farnesene synthase [EC:1.14.15.39 4.2.3.47]
WC1	K11929	outer membrane pore protein E
WC2	K18587	ubiquinone biosynthesis protein COQ9
EDARC0	K22473	alcohol dehydrogenase (quinone), dehydrogenase subunit [EC:1.1.5.5]
	K06390	stage III sporulation protein AA
	K10012	undecaprenyl-phosphate 4-deoxy-4-formamido-L-arabinose transferase [EC:2.4.2.53]
	K03225	type III secretion protein Q
EDARC1	K02777	sugar PTS system EIIA component [EC:2.7.1.-]
EDARC2	K17623	pseudouridine 5′-phosphatase [EC:3.1.3.96]
EDARSTC0	K20449	6-hydroxynicotinate reductase [EC:1.3.7.1]
	K12660	2-dehydro-3-deoxy-L-rhamnonate aldolase [EC:4.1.2.53]
EDARSTC1	K00385	anaerobic sulfite reductase subunit C
	K19309	bacitracin transport system ATP-binding protein
	K06368	response regulator aspartate phosphatase J [EC:3.1.-.-]
EDARSTC2	K09928	uncharacterized protein

## Data Availability

Sequences are deposited on the NCBI repository under the BioProject number PRJNA1303565 (https://www.ncbi.nlm.nih.gov/bioproject/?term=PRJNA1303565).

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
