# Peer review of "Use of Two PGPB Strains for the Valorization of Wastewater Sludge and Formulation of a Biofertilizer for the Recovery of Quercus ilex"

_life, 2025, doi:10.3390/life15091490_

Round 1
Reviewer 1 Report
Comments and Suggestions for Authors
Title:
Use of two PGPB strains for the valorization of waste water sludge and formulation of a biofertilizer for the recovery of Quercus ilex
Recommendation:
Major revision.
Comments:
This study investigates the use of two plant growth-promoting bacterial (PGPB) strains, Bacillus pretiosus and Pseudomonas agronomica, as biofertilizers to enhance the growth of Quercus ilex (holm oak) seedlings. The study evaluates how these bacteria, derived from wastewater sludge, improve plant development, influence soil microbial diversity, and affect antibiotic resistance profiles in the rhizospheric soil. The research combines microbiological, metagenomic, and plant biometric analyses to assess the effectiveness and ecological impact of employing these biofertilizers for sustainable forest management. The subject is relevant and consistent with the aims and scopes of the journal. Some comments and suggestions are offered below with the intent to assist the author in improving the manuscript.
- The numbering of figures and tables in this manuscript has not been checked and is incorrect. The text refers to “Figure 1,” “Figure 2,” and “Table 5,” none of which are present. Please revise thoroughly.
- The references section contains numerous formatting errors and does not adhere to the journal’s guidelines. Please revise thoroughly in accordance with the journal’s style.
- Seedbeds contain 35 alveoli and nine trays were used across the 3×3 (chemical × biological) design, but the manuscript does not clearly define the experimental unit (alveolus, tray, or pooled leachate group), nor how many independent biological replicates were analyzed per treatment. Figures later report n = 3 for nutrition panels, but the plant-level biometrics (Figure 4, p. 8) do not state n or independence. Please specify the blocking (tray) structure, the number of plants analyzed per treatment, and whether trays or plants were treated as replicates in ANOVA to avoid pseudoreplication. A linear mixed-effects model (treatment as fixed; tray/seedbed as random; plant as repeated/within-tray) with multiplicity control (Tukey or FDR) would be more appropriate than Duncan tests.
- Report effect sizes and CIs (not only letters over bars) and state explicitly the n used in Figure 4 (p. 8). The figure shows multiple pairwise letters but lacks sample size and error structure. (Figure 4 images on p. 8.)
- Correct the repeated “five-year” phrasing in preparation/irrigation schedules, which appears to be a translation/typo (likely five-day or similar): biofertilizer “prepared on a five-year basis” and “five-year irrigation periodicity.” Please provide the actual interval and total number of irrigation events.
- Figure numbering: text cites Figure 15 for Bray-Curtis beta diversity, but the embedded figure is Figure 9 (p. 12). Please correct numbering and cross-references.
- The MIC assay uses mixed rhizospheric suspensions plated on Mueller–Hinton with E-test strips at 25 °C; inoculum density is approximated with Uricult (0.5 McFarland). Please justify temperature choice and provide per-antibiotic MIC distributions with statistics (not only PCA with loadings). State replicates per treatment per antibiotic, and how community growth heterogeneity was handled in MIC reading. (Methods and Figure 8, p. 11.)
- Correct “Quercus illex” → ilex and replace literal translations like “vines” with “strains.”
- Spellings: “Kruskal–Wallis,” “AWCD,” and consistent use of decimal separators.
Author Response
- The numbering of figures and tables in this manuscript has not been checked and is incorrect. The text refers to “Figure 1,” “Figure 2,” and “Table 5,” none of which are present. Please revise thoroughly.
Revised and changed.
- The references section contains numerous formatting errors and does not adhere to the journal’s guidelines. Please revise thoroughly in accordance with the journal’s style.
References checked.
- Seedbeds contain 35 alveoli and nine trays were used across the 3×3 (chemical × biological) design, but the manuscript does not clearly define the experimental unit (alveolus, tray, or pooled leachate group), nor how many independent biological replicates were analyzed per treatment. Figures later report n = 3 for nutrition panels, but the plant-level biometrics (Figure 4, p. 8) do not state n or independence. Please specify the blocking (tray) structure, the number of plants analyzed per treatment, and whether trays or plants were treated as replicates in ANOVA to avoid pseudoreplication. A linear mixed-effects model (treatment as fixed; tray/seedbed as random; plant as repeated/within-tray) with multiplicity control (Tukey or FDR) would be more appropriate than Duncan tests.
All the following information has been added to the manuscript.
For biometrics each plant was treated as an individual replicate in each treratment because only one plant was seeded per alveoli and was watering independently.
For nutritional parameters, each treatment was divided in 3 groups of plants to make 3 replicates per treatment.
As for the statistical model used. Due to the number of individuals per treatment and the fact that it is a prospective trial for larger field trials, we consider that an ANOVA and a PCA are sufficient. The ANOVA tells us if the strains perform a promotional function with respect to the controls. And the PCA that indicates that the strain factor has an influence on the data studied. However, we appreciate the advice on the use of linear mixed-effects models and will implement them in future tests.
- Report effect sizes and CIs (not only letters over bars) and state explicitly the n used in Figure 4 (p. 8).
N explicit in text before the figure.
The figure shows multiple pairwise letters but lacks sample size and error structure. (Figure 4 images on p. 8.)
The sample size for the biometric is explained in material and methods section, having 35 individuals for each treatment. Pairwise letters explained in foot figure.
- Correct the repeated “five-year” phrasing in preparation/irrigation schedules, which appears to be a translation/typo (likely five-day or similar): biofertilizer “prepared on a five-year basis” and “five-year irrigation periodicity.” Please provide the actual interval and total number of irrigation events.
That’s a typo. Corrected to “weekly”
- Figure numbering: text cites Figure 15 for Bray-Curtis beta diversity, but the embedded figure is Figure 9 (p. 12). Please correct numbering and cross-references.
Corrected
- The MIC assay uses mixed rhizospheric suspensions plated on Mueller–Hinton with E-test strips at 25 °C; inoculum density is approximated with Uricult (0.5 McFarland). Please justify temperature choice and provide per-antibiotic MIC distributions with statistics (not only PCA with loadings). State replicates per treatment per antibiotic, and how community growth heterogeneity was handled in MIC reading. (Methods and Figure 8, p. 11.)
Sorry for the typo, the correct temperature is 28±2°C which is the standard growth temperature for environmental microorganisms.
Cenoantibiogram is a technique which explores in a prospective way the behaviour of the community. We consider, and looking at the aim of the technique, PCA is the suitable representation of the data and the results obtained to explore this behaviour and its relation with the factors that explain the distribution.
The measurement of the halos is according with the protocol of the bibliography consulted and specified in lines 255-257: “For the quantification of the minimum inhibitory concentration (MIC), the most restrictive halo was used as a reference.”
- Correct “Quercus illex” → ilex and replace literal translations like “vines” with “strains.”
Done
- Spellings: “Kruskal–Wallis,” “AWCD,” and consistent use of decimal separators.
Done
Reviewer 2 Report
Comments and Suggestions for Authors
Review of the article:
‘Use of two PGPB strains for the valorisation of waste water sludge and formulation of a biofertiliser for the recovery of Quercus ilex’ by Vanesa M. Fernández-Pastrana, Daniel González-Reguero, Marina Robas-Mora, Diana Penalba-Iglesias, María José Pozuelo de Felipe, Agustín Probanza, Pedro Jiménez-Gómez
This article discusses the scientifically significant and topical task of using agricultural sludge to produce biofertilisers based on plant growth-promoting bacteria (PGPB), and of assessing their impact on the development of Quercus ilex seedlings. The authors conducted a comprehensive study, assessing plant growth and changes in tissue chemical composition, as well as the structural and functional characteristics of soil microbial communities, using modern molecular and biochemical methods. Demonstrating an interdisciplinary approach, the authors present conclusions of high practical and theoretical value for soil ecobiotechnology and biodiversity. The work's strengths include its high level of relevance, comprehensive analysis and sufficient justification of the research's significance.
Comments on sections
Introduction:
- This section should be expanded to include information on the current views regarding PGPB, including which types of microorganisms are most commonly used in biotechnology, the reasons for their selection and the agroecosystems in which these bacteria are most effective.
- The transition to the topic of antibiotic resistance (pp. 82–83) is rather abrupt and requires logical justification. It would therefore be useful to provide a systematic overview of the known pathways for developing antibiotic resistance among PGPB, as well as a list of the relevant genes and types of microorganisms that possess similar characteristics.
- Some of the literary sources are outdated (e.g. works from the 1990s and earlier) and certain statements (e.g. data on soil loss in Spain) require confirmation using more recent publications.
- The review should be supplemented with current approaches to the biotechnological processing of agricultural sludges, including alternative methods such as pilot or industrial composting.
- The research objective is not clearly formulated at the end of the introduction and must be presented explicitly.
Materials and methods:
- This section does not provide sufficient references to the methods used, making it difficult to reproduce the results.
- Information on the number of repetitions for each experimental variant is lacking in subsection 2.2. The measures taken to ensure sterility when working with soil samples and bacterial inoculants during long-term experiments are not described in sufficient detail. The procedures aimed at preventing cross-contamination need to be more detailed.
- The physical and chemical characteristics of agricultural sludge are given in Section 2.3.2, 'Materials and Methods', but as this material is the result of analysis, it should be accompanied by a description of the methods used.
- The use of the term 'on a five-year basis' in line 2.3.3 raises questions and requires clarification: does this refer to a five-year cycle or other organisational specifics?
- The procedure for selecting rhizosphere soil is not described in sufficient detail in section 2.5.1. The criteria and methodology for sampling should be specified.
- The methodology for determining the transition of the microbial community to the stationary phase of growth (line 227) should be specified in subsection 2.5.2. It is recommended that the corresponding growth curve be added to the appendix.
- The description of collecting and analysing samples for a cenoantibiogram (MIC determination) requires more detailed elaboration. It is necessary to clarify how the influence of non-target microorganisms is excluded, and to justify the choice of specific antibiotics.
- There are inconsistencies in the numbering of figures. For example, why is Figure 3 the first figure in the article? Also, Figure 15 is indicated on line 431, even though there are only 12 figures in the article. Standardisation is required.
Results:
- Figures 4–6 should be accompanied by explanations of the indicated indices (a, b, f, q, m, n, etc.).
- According to Figure 10, the EDARC0 control shows a greater number of Bacillus representatives than the variant treated with the corresponding strain, and the authors should comment on this.
Discussion and conclusions:
- The possible long-term and cumulative effects of biofertiliser application on the soil microbial community, including the potential migration of resistance genes, should be discussed in more detail.
- The limitations of extrapolating the results to other types of ecosystems, soils and agronomic conditions must be clearly stated.
- Lines 677–694 are in Spanish, which is unacceptable — the entire text must be in English.
Technical and stylistic issues:
- There are some typos and incorrect designations (e.g. 'vines' instead of 'strains') in the paper, probably due to machine translation.
- Several sections require linguistic and stylistic editing to improve the clarity and unambiguity of the presentation of the material.
- It is important to clearly standardise the terminology used for 'biofertiliser' and 'organic fertiliser', and to distinguish between their definitions.
Summary
This article makes a significant contribution to the development of soil microbiology and environmentally oriented agricultural technologies. The results obtained may be useful for fundamental and practical research into biosafety and the improvement of the sustainability of agroecosystems. However, the manuscript requires substantial revision, including clarification of methods, detailed statistical analysis, consistent terminology and improved language. A decision on publication can only be made once all comments have been fully and thoroughly addressed.
Comments on the Quality of English LanguageThe article's English is well above average and meets international standards for scientific publications. However, it requires editorial revision to eliminate certain translation, stylistic and terminological shortcomings. While the main ideas and results are presented clearly, the quality of the text will improve once the linguistic issues have been addressed.
Potential reviewers or editors are advised to have the text proofread by a native speaker or professional editor to ensure it is stylistically polished and terminologically accurate.
Author Response
Introduction:
- This section should be expanded to include information on the current views regarding PGPB, including which types of microorganisms are most commonly used in biotechnology, the reasons for their selection and the agroecosystems in which these bacteria are most effective.
Added.
- The transition to the topic of antibiotic resistance (pp. 82–83) is rather abrupt and requires logical justification. It would therefore be useful to provide a systematic overview of the known pathways for developing antibiotic resistance among PGPB, as well as a list of the relevant genes and types of microorganisms that possess similar characteristics.
Some information added. But the principal aim of this work, in the antibiotic statement, is to verify the security and the potential mitigation of use WWTP sludge with the addition of the PGPB in the soils used. That review could be more suitable for a full review work about transmission, resistance and mitigation of antibiotics of PGPB. This is a great idea which we had already discussed in our research group and will certainly carry out in the near future.
- Some of the literary sources are outdated (e.g. works from the 1990s and earlier) and certain statements (e.g. data on soil loss in Spain) require confirmation using more recent publications.
Revised.
- The review should be supplemented with current approaches to the biotechnological processing of agricultural sludges, including alternative methods such as pilot or industrial composting.
The principal aim of this work is to verify the effectivity of the WWTP sludge with the addition of the PGPB. That review could be more suitable for a full review work about biofertilizers and valorised residues, which it’s far from our aim in the present work.
- The research objective is not clearly formulated at the end of the introduction and must be presented explicitly.
Added.
Materials and methods:
- This section does not provide sufficient references to the methods used, making it difficult to reproduce the results.
Revised and corrected
- Information on the number of repetitions for each experimental variant is lacking in subsection 2.2. The measures taken to ensure sterility when working with soil samples and bacterial inoculants during long-term experiments are not described in sufficient detail. The procedures aimed at preventing cross-contamination need to be more detailed.
Changed and information added.
- The physical and chemical characteristics of agricultural sludge are given in Section 2.3.2, 'Materials and Methods', but as this material is the result of analysis, it should be accompanied by a description of the methods used.
Added.
- The use of the term 'on a five-year basis' in line 2.3.3 raises questions and requires clarification: does this refer to a five-year cycle or other organisational specifics?
Changed in text, was a typo for “weekly watered”.
- The procedure for selecting rhizosphere soil is not described in sufficient detail in section 2.5.1. The criteria and methodology for sampling should be specified.
Information added.
- The methodology for determining the transition of the microbial community to the stationary phase of growth (line 227) should be specified in subsection 2.5.2. It is recommended that the corresponding growth curve be added to the appendix.
Added.
- The description of collecting and analysing samples for a cenoantibiogram (MIC determination) requires more detailed elaboration. It is necessary to clarify how the influence of non-target microorganisms is excluded, and to justify the choice of specific antibiotics.
Information added.
- There are inconsistencies in the numbering of figures. For example, why is Figure 3 the first figure in the article? Also, Figure 15 is indicated on line 431, even though there are only 12 figures in the article. Standardisation is required.
Changed and corrected throughout the manuscript.
Results:
- Figures 4–6 should be accompanied by explanations of the indicated indices (a, b, f, q, m, n, etc.).
Done
- According to Figure 10, the EDARC0 control shows a greater number of Bacillus representatives than the variant treated with the corresponding strain, and the authors should comment on this.
The limitation of the technique conditions the accurate count of mo, for this reason we do not find that there is a significant difference between the result of EDARC0 and EDARC1.
Discussion and conclusions:
- The possible long-term and cumulative effects of biofertiliser application on the soil microbial community, including the potential migration of resistance genes, should be discussed in more detail.
They are mineralizing bacteria that have a limited relationship with the plant model used and therefore we interpret that at the end of the fertigation treatment the presence of the added bacteria disappears or may remain in a very residual way. However, as indicated in the manuscript, longer trials are needed to confirm the results obtained.
- The limitations of extrapolating the results to other types of ecosystems, soils and agronomic conditions must be clearly stated.
Information added.
- Lines 677–694 are in Spanish, which is unacceptable — the entire text must be in English.
Sorry for the mistake. Corrected and revised.
Technical and stylistic issues:
- There are some typos and incorrect designations (e.g. 'vines' instead of 'strains') in the paper, probably due to machine translation.
Corrected
- Several sections require linguistic and stylistic editing to improve the clarity and unambiguity of the presentation of the material.
Revised.
- It is important to clearly standardise the terminology used for 'biofertiliser' and 'organic fertiliser', and to distinguish between their definitions.
Revised.
Reviewer 3 Report
Comments and Suggestions for Authors
The manuscript under review contains interesting results of research on the impact of two strains of plant growth-promoting bacteria (PGPB), Bacillus pretiosus and Pseudomonas agronomica, on holm oak (Quercus ilex) seedlings. In my opinion, the manuscript should be supplemented with the authors' indication of the possibility of using the obtained research results in agroforestry systems. What difficulties might be associated with this? What problems still need to be solved?
Comments
Introduction
It introduces the reader to the research problem well.
Materials and Methods
Please provide information on where the research was conducted and during what period. When was the experiment set up? When was the data collected? When were the analyses performed?
Results
The description of the results is complete and detailed. The figures are clear and necessary. The same applies to the tables.
Please do not start a sentence with „Figure 4 shows”.
References
There are a large number of publications (68). Some of them are quite old, and some were published in the last century. I advise you to focus only on the most recent publications.
Author Response
Introduction
It introduces the reader to the research problem well.
Materials and Methods
Please provide information on where the research was conducted and during what period.
As appears in section 2.3.4., the experiment was developed in a phytotron in our facilities. Added information in the same section.
When was the experiment set up? During September of the same year (2023)
When was the data collected? Just at the end of the experiment in October 2024.
When were the analyses performed? Starts in October and ends in November (2024).
Results
The description of the results is complete and detailed. The figures are clear and necessary. The same applies to the tables.
Please do not start a sentence with „Figure 4 shows”. Done
References
There are a large number of publications (68). Some of them are quite old, and some were published in the last century. I advise you to focus only on the most recent publications.
Thanks for the advice, but after revising it, some of the old references refers to classical technics, protocols or definitions.
Round 2
Reviewer 1 Report
Comments and Suggestions for Authors
Title:
Use of two PGPB strains for the valorization of waste water sludge and formulation of a biofertilizer for the recovery of Quercus ilex
Recommendation:
Accept in current form
Comments:
After the previous review, the manuscript has been revised and modified quite well. Therefore, I suggest that this manuscript can be accepted in current form.
Reviewer 2 Report
Comments and Suggestions for Authors
The article should be accepted in its present form.